# Patient and Clinician Preferences for Genetic and Genomic Testing in Non-Small Cell Lung Cancer: A Discrete Choice Experiment

**DOI:** 10.3390/jpm12060879

**Published:** 2022-05-26

**Authors:** Simon Fifer, Robyn Ordman, Lisa Briggs, Andrea Cowley

**Affiliations:** 1Community and Patient Preference Research Pty Ltd., Sydney, NSW 2000, Australia; ordman.robyn@gmail.com; 2Thoracic Oncology Group Australasia, Sydney, NSW 2000, Australia; drlisabriggs@gmail.com; 3Rare Cancers Australia, Sydney, NSW 2000, Australia; 4Roche Products Pty Limited, Sydney, NSW 2000, Australia; andrea.cowley@roche.com

**Keywords:** NSCLC, genetic testing, genomic testing, patient preference, discrete choice experiment

## Abstract

Precision (personalised) medicine for non-small cell lung cancer (NSCLC) adopts a molecularly guided approach. Standard-of-care testing in Australia is via sequential single-gene testing which is inefficient and leads to tissue exhaustion. The purpose of this study was to understand preferences around genetic and genomic testing in locally advanced or metastatic NSCLC. A discrete choice experiment (DCE) was conducted in patients with NSCLC (*n* = 45) and physicians (*n* = 44). Attributes for the DCE were developed based on qualitative interviews, literature reviews and expert opinion. DCE data were modelled using a mixed multinomial logit model (MMNL). The results showed that the most important attribute for patients and clinicians was the likelihood of an actionable test, followed by the cost. Patients significantly preferred tests with a possibility for reporting on germline findings over those without (β = 0.4626) and those that required no further procedures over tests that required re-biopsy (β = 0.5523). Physician preferences were similar (β = 0.2758 and β = 0.857, respectively). Overall, there was a strong preference for genomic tests that have attribute profiles reflective of comprehensive genomic profiling (CGP) and whole exome sequencing (WES)/whole genome sequencing (WGS), irrespective of high costs. Participants preferred tests that provided actionable outcomes, were affordable, timely, and negated the need for additional biopsy.

## 1. Introduction

Lung cancer is the leading cause of both cancer related mortality and burden of disease in Australia with a 5-yr survival rate of 20% [1]. It represents the fifth most commonly diagnosed cancer with 45.8% of people diagnosed at the most advanced stage of the disease when it is incurable and significantly impacts survival [2]. The rates of diagnosis at a locally advanced or advanced stage in Aboriginal and Torres Strait Islander people is 59% [3]. Therefore, it is important to ensure the early diagnosis and treatment of patients.

In recent years, there has been a paradigm shift in cancer treatment towards precision (personalised) medicine, which adopts a molecularly guided or targeted approach to therapy. Clinical practice guidelines recommend targeted therapy for patients with advanced or metastatic lung cancer whose tumour expresses specific genetic/genomic mutations [4,5]. When patients undergo genomic testing, identified alterations can determine which patients are most likely to benefit from certain treatments. Conversely, it may identify therapies that are unlikely to be of benefit, thereby minimising unnecessary treatment [4]. Targeted agents for treating lung cancer first emerged in Australia with the approval of gefitinib, the endothelial growth factor receptor (EGFR) inhibitor in 2003. This was followed by approvals for erlotinib (another EGFR inhibitor) in 2006 and crizotinib, the first anaplastic lymphoma kinase tyrosine kinase inhibitor (ALK TKI) agent, which was TGA-approved in 2011. Thus, when targeted treatment was first approved, only the routine testing of *EGFR* and *ALK* were recommended, despite there being other actionable genomic alterations identified [6,7]. In Australia, the current firstst line TKI treatments for EGFR are erlotinib, gefitinib and afatinib (an osimertinib for EGFR T790M-positive); for ALK they are alectinib ceritinib or crizotinib; and for ROS1 the treatment is crizotinib [8,9,10]. The number of genetic and genomic drivers and targeted therapies approved for use in NSCLC are growing quickly, with an even greater number currently under investigation [8,11].

Current standard of care diagnostic testing for NSCLC in Australia includes sequential single-gene testing for *EGFR*, *ALK* and *ROS1* which are prioritised due to their population prevalence and subsidised treatments [9,10,12]. Unfortunately, this approach has several disadvantages, including testing inefficiencies and tissue exhaustion, resulting in high rates of repeat biopsy (approximately 30%) [13,14]. Given the rapid and exponential emergence of new clinically relevant genomic alterations and associated targeted therapies, there is a substantial unmet need for a more comprehensive and efficient approach to identify genetic and genomic alterations. Next generation sequencing (NGS) technology facilitates the rapid and comprehensive identification of genomic alterations by allowing large stretches of tumour DNA to be sequenced, thus ensuring that patients can access appropriate targeted therapies in a timely manner. Comprehensive genomic profiling (CGP) is an NGS sequencing approach that detects novel and known variants of the four main classes of genomic alterations (single nucleotide variants [SNVs]; copy number alterations [CNAs]; insertions and deletions [indels]; and rearrangements) and genomic signatures (tumour mutation burden [TMB] and microsatellite instability [MSI]) to provide prognostic, diagnostic and predictive insights that inform research or treatment decisions for individual patients across all cancer types. CGP, therefore, consolidates multiple molecular biomarker tests into a single assay and minimises the need for sequential biomarker tests. This may lead to a reduction in the number of repeat biopsies that are required after the initial sample is depleted.

Current international guidelines recommend testing to be performed using a broad, panel-based NGS driven approach [4,15]. Despite these international recommendations, the use of sequential single-gene tests for lung cancer molecular profiling remains common in Australia [16]. The biggest barrier to more comprehensive testing is cost, and in many countries the cost of testing falls either to the patient or to the institution/hospital [16]. Unfortunately, this is still the case for patients with NSCLC in Australia, with NGS remaining unfunded, despite there being a clear benefit for both the clinician and patient [4,15]. Typically, single-gene tests have been approved by the Medical Services Advisory Committee (MSAC) as they have been designed to establish eligibility for subsidised treatments (so-called co-dependent technologies) [17]. Since 2015, MSAC has emphasised the clinical utility of the test, with an importance placed on the test producing an actionable result, that is, one which leads to changes in health decisions and consequent treatment outcomes for patients [17].

The current health-political environment, as highlighted by recent reviews including the Australian House of Representatives Parliamentary inquiry on approval processes for new drugs and novel medical technologies in Australia [18], the MSAC guidelines review [19], and the NHS genomics implementation plan [20], has recognised the importance of patient and physician preferences when implementing new technologies into clinical care. Assessing stakeholder preferences through choice tasks as opposed to simple questionnaires poses benefits due to the complexities associated with healthcare decision-making. Observing patterns of behaviour using forced-choice trade-offs offers insight into real-world patterns of behaviour, allowing a greater understanding of the value of certain features of a given product or technology in a controlled, hypothetical environment. For these reasons, patient and physician preferences were assessed using a choice task to determine elicited preferences, as opposed to a simple questionnaire design.

Although preference research among NSCLC treatments and outcomes exists in the literature, little is understood about the preferences for features of genetic and genomic tests for this cohort. The purpose of the present study was to better understand patient and physician preferences around genetic and genomic testing for people living with locally advanced or metastatic NSCLC in the Australian context.

## 2. Materials and Methods

This study was conducted in two parts. Figure 1 provides an overview of the study design. Firstly, a review of the literature of current testing procedures that are available in Australia was conducted to inform in-depth interviews with patients and clinicians. These interviews were conducted with people diagnosed with NSCLC, oncologists, respiratory physicians and pathologists and informed the development of the second part of the research, a quantitative survey incorporating a discrete choice experiment (DCE). The interviews and DCE are described more fully below. This study was conducted in full conformance with the Guidelines for Good Pharmacoepidemiology Practice published by the International Society of Pharmacoepidemiology (ISPE) [21] and local laws and regulations and codes of research conduct. It received Human Research Ethics Committee approval from Bellberry (approval number HREC 2020-03-252). All participants provided written, informed consent.

### 2.1. Interviews

#### 2.1.1. Interview Participants

Interviews were conducted with patients with NSCLC and clinicians (physicians and pathologists). Eligible participants included patients who had been diagnosed with locally advanced (stage IIIb) or metastatic (stage IV) lung cancer aged 18 years or over; physicians (oncologists and respiratory physicians) who had treated a minimum of five patients with stage IIIb or IV NSCLC; and pathologists who had ordered genetic and genomic testing (either by single-gene tests or NGS assays) for at least five people diagnosed with stage IIIb or stage IV NSCLC by a medical professional. Patients were recruited via the Lung Foundation Australia and community engagement; physicians via invitation from a third-party marketing research services company (EKas); and pathologists via a Roche contact list. The participants were compensated for their time and contribution (AUD 100 for patients; AUD 400 for clinicians) either by direct deposit, or as a contribution to Lung Foundation Australia.

#### 2.1.2. Interview Approach

Participants underwent online video conference interviews which lasted from 45 to 60 min. The interviews explored participant awareness of and use of genetic and genomic testing in Australia and determined the key attributes that were of value to them when selecting a genetic or genomic test. A preference for tissue or liquid, and single-gene versus NGS options (small and medium panels, CGP and whole exome sequencing (WES) or whole genome sequencing (WGS)) was explored. Interview questions are included in the Appendix A (Appendix A). All participants provided written and verbal informed consent.

#### 2.1.3. Data Analysis

All interviews were audio recorded and transcribed. Interview data including transcriptions and analysis were managed in a specialised qualitative research software (NVivo). Once familiarised with the interview contents, broad themes and subsequent codes developed a preliminary code frame. The interview contents underwent a thematic analysis using this code frame. Finalisation of the code frame and interview contents was reached upon discussion between the two coders working independently.

### 2.2. Discrete Choice Experiment

This study used a DCE approach to gain an understanding of the genetic/genomic testing preferences of patients with locally advanced (stage IIIb) or metastatic (stage IV) lung cancer (in line with clinical practice guideline recommendations for eligibility for genomic testing) and physicians (oncologists and respiratory physicians) who treat lung cancer patients. After careful consideration and input from desk research, expert industry opinion, and interview feedback, both oncologists and respiratory physicians were included in the sample as both hold some responsibility for ordering genomic tests in the relevant patient population. DCEs have a firm theoretical background that is grounded in psychology and economics and they are now commonly used in health to understand preferences [22,23,24,25]. DCEs utilise a survey (questionnaire) that presents participants with scenarios and asks them to choose an alternative within each scenario that maximises their utility (i.e., satisfaction), according to their own value framework. DCEs are primarily used to model the trade-offs and preferences revealed by the choices that people make. The attributes that were used in the DCE were developed through the formative research described above (literature review and in-depth interviews) and discussion among the author team. The attributes of interest were tissue requirements (blood test or lung tissue biopsy), time between test and results, cost of test, chance of an actionable outcome, funding, number of genes tested, location of process and analysis, whether germline findings would be reported, and reporting capabilities (see Table 1). Attribute levels were chosen to reflect current genetic and genomic testing capabilities, and possible future methods.

The study and experimental designs followed good practice guidelines [26,27].

#### 2.2.1. Survey Participants

Patients were eligible to participate if they had stage IIIb or IV NSCLC, were aged 18 years or over and were fluent in English. Clinicians were eligible to participate if they had treated a minimum of five patients with stage IIIb or IV NSCLC and were fluent in English. Pathologists were not included in the quantitative (DCE) part of this study as they are not involved in the referral decision-making process for genetic or genomic testing. All participants provided written informed consent.

Convenience sampling was used to identify potential participants. Participants opted-in to the survey after hearing about it from the Lung Foundation Australia via email or social media, through third-party marketing research services companies (Pure Profile, Dynata, Stable research, Lucid or EKas), or by community recruitment through social media. Participants were compensated for their time: AUD 30 (patients) or AUD 140 (clinicians) for their participation.

#### 2.2.2. Survey Instrument

The survey instrument consisted of a section for the DCE task (Figure 2), as well as non-experiment questionnaire sections assessing participant characteristics, test and treatment history, and background knowledge and understanding of genomics in cancer care. For the DCE section, a series of discrete choice scenarios focusing on the clinical and logistical attributes of genetic and genomic testing were presented to patients and physicians. The survey was piloted with eleven patients and ten physicians, and minor adjustments to the design were made before launching broadly. These changes included simplification of the scenarios presented to participants. The final attributes included in the DCE are listed in Table 1.

Survey completion took approximately 20 to 25 min for patients and 15 to 20 min for clinicians. The clinician DCE contained one additional attribute on reporting and interpretation. The participants were provided with a brief explanation on genetic and genomic testing and were instructed to imagine that their healthcare professional had advised that a further genetic and genomic test was required, and to choose between the hypothetical options presented. The options included two test alternatives and one opt-out. The opt-out was represented by the current single-gene testing paradigm.

Seventy-two DCE scenarios were split into six blocks so that each participant was presented with 14 discrete choice scenarios in which respondents were asked to choose their preferred option for each scenario. Of the 14 discrete choice scenarios presented, 12 were included in the DCE design and analysis to allow for participant task familiarisation and additional data consistency checks. All attributes and levels were explained in full. Understanding checks were performed both before and after the DCE scenario; for example, using knowledge-check questions presented with context cues in the survey (e.g., “if you consider the following two test options, which one would be more likely to provide you with actionable information”) and self-report questions (e.g., “please rate your understanding of the scenarios you have just completed”). By observing how respondents changed their responses to the various situations posed, the importance they placed on the presented attributes could be modelled. An example DCE scenario is shown in Figure 1.

### 2.3. Data Analysis

Combinations of the levels presented were designed using a Bayesian D-efficient design with naïve priors to account for level direction in an Ngene v1.1.2 (ChoiceMetrics, Sydney, NSW, Australia) [28]. The combinations of levels presented in the DCE instrument use a D-efficient design structure that uses naïve priors to account for the sign of the parameters and level order in Ngene [28]. The relative attribute importance was calculated by comparing the change in utility from the lowest level to the highest level for each parameter. In this case, because the attributes were coded as categorical variables using effects coding [29], the magnitude of the parameters could be compared directly to provide a measure of relative importance. As with most DCEs, an a priori sample size calculation was not performed. Data from patients who finished the survey too quickly, gave nonsensical answers, were suspected duplicates or who admitted to a poor understanding of the DCE (a rating of less than six on a scale from 0 (‘did not understand at all’) to 10 (‘understood perfectly’) were removed from the analysis.

Demographic characteristics and treatment history were summarised descriptively using an SPSS v25 (IBM, New York, NY, USA). The DCE data were modelled using a mixed multinomial logit model (MMNL). The MMNL structure allows greater flexibility in the model coefficients so that it can account for preference heterogeneity between participants and relax the restrictive assumptions of the standard multinomial logit model (MNL). It achieves this additional level of accuracy by allowing the parameter coefficients to be random variables drawn from a pre-specified distribution, (ө). This means that the set of parameter estimates may vary between each individual participant such that the model definition includes a 𝛽𝑛𝑗𝑘 which is specific to participant 𝑛.

Point-estimates for the parameter coefficients were obtained by estimating the mean value over the *n*-sets of participant-specific coefficients. The relative importance of each attribute was calculated by finding the maximum difference in utility between the attribute’s levels and expressing it as a percentage of the sum of all maximum differences.

The statistical significance was determined by the *p* < 0.05 criteria, and the adjusted McFadden Pseudo R-squared which accounts for the number of parameters in a model was used to assess model fit. Statistical analyses were performed using an Nlogit version 6 (Econometric Software, Inc., Plainview, NY, USA).

## 3. Results

### 3.1. Interviews

#### 3.1.1. Participants

The interviews were conducted between December 2018 and February 2019 and included 15 participants (six patients, seven physicians and two pathologists). The mean age of patients was 48.5 ± 9.0 years (range 38 to 50 years); 5/6 (83%) were female, 5/6 (83%) had stage IV NSCLC, three (50%) had mutations in *EGFR* and three (50%) had mutations in *ALK*. Physicians were either medical oncologists (4/7, 57%) or respiratory physicians (3/7, 43%), mostly practicing in metropolitan areas (6/7, 86%). Two pathologists were interviewed and both worked in the public hospital setting.

#### 3.1.2. Patients

Patients identified the time to diagnosis as being a significant problem, with some experiencing symptoms for an extended period of time. Patients noted that little discussion around testing occurred despite the majority having undergone genetic and genomic testing using a tissue biopsy. Patients also recognised that access to full genomic sequencing may be limited.
Patients cannot find opportunities to have their, you know, full genomic sequencing performed, pay for it if they wish, or just be clear with how much it costs, where to go and how long it is going to take. They’re the three things people want to know and yet to find that information is like finding a needle in a haystack, and if you don’t know … who’s who in the zoo then you miss out’. (Patient 4, Female, VIC)

Patients reported wishing to have access to treatment opportunities that would be available through CGP.
You just offer so many more opportunities to patients to access treatments that are more personalised and targeted and you know provide them with really good quality of life’. (Patient 4, Female, VIC)

There was also a willingness to pay for this information, although the cost of treatment was a consideration for some patients.

Patients reported wishing to have access to the tests yielding the greatest amount of information.
The more I know the better I equip myself with information and make better decisions. I don’t see any disadvantage in more knowledge yeah…the more you know the more information you know about your genes the better’. (Patient 3, Female, QLD)
I would go for the more detailed one, you would be able to have more of a chance of knowing what kind of cancer it mutates to which I don’t have, and then you’ll have the best chance of finding a treatment that suits you’. (Patient 6, Female, VIC)
It is only through genomic testing you know real sequencing of the cancer that has allowed them to take you know a combo of drugs. The patient demand is there for it, it is seen as a positive not a negative’. (Patient 2, Male, VIC)

#### 3.1.3. Physicians

Physicians echoed patient sentiments, highlighting the importance of more comprehensive tests such as CGP in driving clinical decision-making, and the resulting potential to improve patient outcomes and quality of life. Although the cost of the test itself was not a limiting barrier to recommending a test, the potential financial burden to the patient in the event that they were matched to an unlisted and unfunded treatment was a concern to the physicians. Although the accuracy of testing was often not mentioned forthright, it was likely assumed that all tests available had been appropriately validated and approved. Physicians were concerned about the timeliness of tests, and while location of testing was not of high importance, they did feel this was a driver for the timeliness of reporting.
I think that’s what patients want because they want a road map of where to go from here rather than every so often saying, “Oh well we could do this other test and this might show this, and if we found that then we could do that’’. (Physician 2, Medical Oncologist, NSW)

#### 3.1.4. Pathologists

Pathologists reported that their primary focus was on tissue preservation and conservation during processing in the lab, compared to the clinic or interacting with patients. While pathologists do not provide recommendations to physicians on which tests to run, they are involved in providing advice on which treatment to consider. Their role in providing an interpretation of the genomic results included participation in multidisciplinary team meetings. They noted that the need to retest was common due to tissue insufficiency or the requirements for fluorescent in situ hybridisation (FISH) and immunohistochemistry (IHC) to access treatments. However, they felt the current sequential approach to testing was flawed.
Well, most molecular testing on cancer is one test for one gene for one drug, and you can have a patient centred approach where this is the patient they have a disease we test for all genes and if a gene has a mutation then they get a particular drug. The co-dependent methodology is fatally flawed in my opinion’ (Pathologist 1, Male, NSW)

#### 3.1.5. Attributes Identified

A list of attributes that were identified during the qualitative process are listed in the Appendix A (Appendix A). These attributes were further refined before being used in the DCE (see Table 1).

### 3.2. Results of the DCE

Surveys were conducted in Australia (NSW, QLD, VIC, WA, SA) across metropolitan, regional and rural areas between October 2020 and May 2021. Overall, 48 patients and 44 clinicians completed the DCE. Of these, 45 patient and 44 clinician responses were included (three patient responses were excluded because they completed the survey too quickly, provided answers that did not fit, were suspected duplicates or admitted to a poor understanding of the DCE). The demographics of participants are reported in Table 2. Of the 44 clinicians, 23 were medical oncologists and 21 were respiratory physicians.

#### 3.2.1. Mixed Multinomial Logit Model Results for Patients

The model output for the 45 included patients is reported in Table 3. Patients displayed a significant preference for tests that required no further procedures over tests that require a re-biopsy (β = 0.5523), as well as those with a possibility for germline findings (i.e., inherited genetic findings) over those without (β = 0.4626). The significantly negative cost parameter (β = −0.0004) indicates a decrease in preference with an increase in out-of-pocket costs. Holding all else equal, patients preferred CGP (β = 0.2499), then whole exome sequencing/whole genome sequencing (WES/WGS) (β = 0.2198), then medium gene panels (β = 0.1311), over a small gene panel.

#### 3.2.2. Mixed Multinomial Logit Model Results for Physicians

The model estimates for the 44 included clinicians are reported in Table 4. Physicians mirrored the sentiment of patients regarding the likelihood of an actionable outcome and whether the actionable outcome (e.g., treatment) would be funded, with a likelihood of 75% and no out-of-pocket costs preferred to a <1% likelihood with out-of-pocket costs (𝛽𝐴_𝐶__75__𝐹_ = 5.1731). In the physician sample a test with no additional procedures was preferred to one that required a re-biopsy (𝛽_𝑇𝑁_ = 0.857). A test turnaround time of less than 2 weeks was significantly preferred to 6–12 weeks (𝛽_𝑇𝑀𝐸__2_ = 1.0958). Tests with germline findings were preferred to those without (𝛽_𝐺𝐸𝑅𝑀_ = 0.2758) and a test sample processed and analysed in Australia was preferred to overseas (𝛽_𝐿𝑂𝐶_ = 0.1712).

In terms of interpretation and reporting, compared to reporting on genetic mutation status only, reports that also examined the eligibility for established or under-investigation targeted therapies and enabled access to a local decision support service were significantly preferred (𝛽_𝑅𝐸𝑃𝐷𝑆𝑆_ = 0.6043). As observed with patients, the significantly negative cost parameter (𝛽_𝐶𝑂𝑆𝑇_ = −0.0006) indicates a decrease in preference with an increase in out-of-pocket costs.

### 3.3. Attribute Importance

The most important attribute for patients was the likelihood of an actionable outcome resulting from the test, and whether this outcome would be funded (Figure 3). The second most important attribute was the cost of the test, followed by the time between the test and the results. The remaining attributes: tissue requirements, location of processing and analysis and number of genes tested, each accounted for <10% of decisions individually.

As with the patient preference, the dominant clinician attribute was the likelihood of an actionable outcome resulting from the test and whether this actionable outcome (e.g., treatment) would be funded, accounting for 45% of decision-making. The cost (17%) and number of genes tested (13%) attributes were the only remaining attributes that contributed more than 10% of decision-making individually (Figure 3).

### 3.4. Preference Share

The model results can be used to estimate preference share for certain tests. For patients, CGP was preferred even over WES/WGS in all tissue requirement comparisons. When all treatments were funded, WES/WGS and CGP were equally preferred among physicians. However, CGP was substantially preferred over WES/WGS when treatments were not funded. When treatments were not funded, physician preferences were heavily impacted by any additional procedures, with the preference share decreasing more dramatically with tumour re-biopsy compared to a blood test (Appendix A).

When all treatment was funded, the physician uptake was higher than the patient uptake when a re-biopsy of tumour tissue was required. When all treatment was not funded, there was little impact on patient uptake but a significant reduction in physician uptake (compared to funded treatments) (Table 5).

## 4. Discussion

Preference research to date has focused primarily on the patient preferences around treatments and treatment outcomes. This study is the first to report the genetic and genomic testing preferences among NSCLC patients living in Australia, and the clinicians providing their care. The study showed that patients valued testing that could use existing lung tissue (compared to re-biopsy), took less than 2 weeks to obtain results, had actionable results, were processed in Australia and could report on germline findings. Clinicians held similar views, although they also preferred tests that could assess larger numbers of genes. This could be due to insufficient information and resources currently available to patients regarding the benefits of larger gene panels. Clinicians also indicated they preferred reports on genetic mutation status and eligibility for targeted therapies, including those that were available in Australia without cost to the patient, either via the Pharmaceutical Benefits Scheme (PBS) or under investigation via clinical trial.

For both patients and clinicians, the most important attribute was the likelihood of an actionable outcome, followed by cost. However, it is important to note the complex linkages between actionability and cost. The lack of an actionable outcome may occur due to the absence of an accessible targeted therapy, but could also be due to test failure and/or insufficient tissue. Inaction due to test failure has downstream consequences on overall cost, due to the need for re-biopsy, the potential for ineffective treatments, paying for continued sequential tests, etc. Aside from cost, there was also a strong preference to avoid lung-tissue re-biopsy, though the current Australian standard of care involves sequential single-gene tissue testing for mutations in *EGFR* (using PCR), followed by *ALK* and then *ROS1* (using triage IHC testing and confirmatory FISH). This approach is inefficient and results in tissue exhaustion and often leads to multiple re-biopsies to complete testing [30,31,32]. The rapid and exponential emergence of new clinically relevant genomic alterations and associated targeted therapies means that a sequential single-gene testing approach is quickly becoming redundant. In addition, initial testing may lead to a non-actionable outcome, requiring approximately one-third of patients to undergo more testing, further delaying treatment initiation, or resulting in exposure to higher toxicities or less efficacious treatments for longer periods of time. Therefore, there is a substantial unmet need for a more comprehensive and efficient approach to identify genetic and genomic alterations.

CGP facilitates the early identification of actionable genomic alterations and appropriate treatment selection while reducing the need for re-biopsy, thus improving outcomes and quality of life in people with NSCLC [33,34]. This is particularly important given the large proportion of patients who are diagnosed with NSCLC at the most advanced stages [3]. CGP/WES/WGS/larger NGS panels can also identify co-occurring genetic alterations to better define genomic complexity and prognosis in patients with NSCLC [35]. This is particularly important for patients who have undergone multiple lines of treatment and have developed multiple resistance mutations as well as providing opportunities to access clinical trials [33]. Despite this, significant barriers in the uptake of CGP have been reported in the Australian setting [36] and depend highly on local expertise and resources. In addition, the PBS does not currently recognise liquid biopsy (plasma/CSF) testing in NSCLC to access targeted therapies. This is problematic in circumstances where the tissue sample is insufficient in either quantity or quality, when the tumour tissue is unable to be directly accessed due to its location, or when the patient is physically unable to have a tissue biopsy procedure performed. In cases where access and tissue quantity or quality is an issue, there is a strong preference for WES/WGS and CGP that uses liquid biopsy samples (i.e., plasma/blood) to ensure equitable access is provided to targeted treatments for all patients.

The use of CGP in treatment selection is not without ethical dilemmas [37]. In NSCLC, this may mean that genetic and genomic tests identify mutations that can only be treated with agents outside of the reimbursed setting—and potentially at significant financial cost to patients. Others have reported that patients perceive disparity in access based on a lack of public funding for testing as an issue [37]. Cost is a key attribute in the decision-making process for testing as it highlights issues of affordability and equity of access, as well as physician perceptions of patients’ abilities to pay for testing. For some patients, cost may not be a barrier for testing but may lead to clinicians not offering testing because of assumptions made about the patient’s ability to pay. Conversations around cost can be uncomfortable, particularly if the results of the test may reveal an outcome which is inaccessible and/or unaffordable.

Further, patients preferred testing with shorter turnaround times, an attribute which has been reported in other DCE of CGP [36]. CGP may also detect co-mutations and may identify multiple targeted therapies that the patient is eligible for. However, the current funding system limits access to targeted therapies as monotherapy only [38].

While clinical utility and actionable outcomes are key attributes of importance for patients and clinicians, our study highlights that there is also a strong preference for testing that may report germline findings. Clinicians preferred tests that could test larger numbers of genes and reports on genetic mutation status and eligibility for targeted therapies that were established or under investigation, including access to a local decision support service. Our findings are consistent with the study by Best et al. (2020) that highlights a high proportion (64%) of patients would still elect to receive results, even when there was only a slight chance that it would lead to an actionable outcome [39]. Tests that report findings that may benefit others are also highly preferable. Therefore, there is significant value in knowing the germline and non-actionable findings and outcomes, highlighting the need for genetic counselling and evaluation of the patient and their relatives for germline mutations that may result in any increased risk of cancer or other conditions.

While patients reported a willingness to pay for more comprehensive testing, that willingness decreased where the resultant treatment choice was not funded. Cost, in terms of monetary outlay and turnaround time, remains a large barrier to the adoption of WES/WGS even in cases where there are no additional tissue requirements. A high chance of out-of-pocket treatment costs (i.e., not funded) drastically impacted patient preference share for WES/WGS. Conversely, the funding of treatment had little impact on the preference for CGP, regardless of tissue requirements. For both WES/WGS and CGP, additional tissue requirements decreased the patient preference share, though less-so when a liquid biopsy was required (i.e., plasma), compared to a tumour re-biopsy. Interestingly, CGP was preferred even over WES/WGS in all tissue requirement comparisons. Cost remains a significant consideration across all cancer care, which may account for tissue requirements, the location of processing and analysis, and the number of genes tested. Each of these factors accounted for < 10% of patient decisions individually.

The impact of cost on decision-making highlights the need for public funding of not only the test itself, but also of the proposed treatment(s). While a minority of patients have a willingness to pay, there are benefits to all public taxpayers for these technologies to be funded and thus provide equitable access to genomics to derive improvements in population outcomes. From a payer perspective, the use of NGS panels is less expensive, more reliable, and requires less tissue samples than sequential testing in NSCLC [40,41]. In an Italian study, costs were EUR 1375/patient for sequential testing compared to EUR 770/patient for NGS [40], and significant savings for payers have also been reported [41]. In addition to the cost benefit, NGS offers wider molecular characterisation, and therefore, better patient classification (and may reduce the use of expensive ineffective standard treatments); or example, in the case of *KRAS* positivity [35], it is the difference between being able to access a targeted treatment of chemotherapy, resulting in a substantial difference in both quality of life and overall outcomes. Due to the rapid evolution of genomic markers that have been identified over recent years, many more targeted NSCLC treatments are likely to be registered over the next two to three years, not only increasing the need for more comprehensive molecular testing approaches in the immediate future, but also inflating their potential value [35]. Funding should also consider access to genetic counselling, especially where limited treatment choices are available [35].

Physicians noted that they wished to have appropriate reports that also examined the eligibility for established or under-investigation targeted therapies and enabled access to local decision support services. Further, while pathologists do not appear to play a role in the decision of which test to undertake, they play a key role in the interpretation and reporting of NGS tests. Ideally the biologic, oncogenic relevance and prognostic or predictive significance of molecular alterations identified should be standardised through comparison to a precision oncology database, and overseen by a multidisciplinary team ‘tumour board’ [35]. To date, however, no global consensus has been adopted between different local and national molecular tumour boards [35]. This has also been reported by others [42], reinforcing the need to improve clinician confidence in communicating results to patients and in selecting appropriate treatments based on genetic and genomic testing results.

Limitations and areas of future research: Our DCE study relied heavily on qualitative formative research, and as such, assumed that all relevant attributes had been captured and included in the DCE. It is possible that not all attributes were included, especially given the small sample size of our initial interviews. Further, careful consideration of the DCE design was necessary to ensure that only the most important attributes were captured. We believe that the combination of literature review, patient interviews, expert (physician) opinion and feedback from authors ensured that this was the case. Other potential biases around sampling included selection bias, since the participants who chose to be involved via community-based and social media advertising may reflect a group that is generally more engaged and thus less ambivalent; more experienced; and more informed about testing and treatment options. Selection bias is reflected in the average age of participants being younger than that of the population of NSCLC patients in Australia. In addition to this, health status information was not collected, and so sample representativeness cannot be assured to the broader Australian population with NSCLC. The design of choice tasks such as DCE accounts for some variables that tend to have a greater impact on stated preference surveys, such as socioeconomic demographics and prior knowledge. The inclusion of background explanation and attributes accounting for financial influences serve to mitigate some of the traditional issues with sample representativeness.

As noted, the participants were provided a full background and explanation on all attributes and levels of the task, and so it is possible that their decisions may not be representative of the real-world setting. However, a shift towards patient empowerment over recent years includes the development of consumer education resources for products and technologies. In this way, the setting in which patients are briefed prior to making decisions may be *more* representative of the cohort of the near future.

Prior findings have indicated that patients presented with larger gene panels first may inflate preference scores [43], although this was minimised by the randomisation of the presented scenarios. Finally, the sample was too small to perform a segmentation analysis, although the usefulness of this in the absence of understanding the proportions of segments in the population would be of limited value.

Future research should determine the preferences around the risk profiles that are associated with various biopsy methodologies for testing and determine how this can be balanced against the benefits of genomic testing in terms of actionability and access to funded or unfunded treatments.

## 5. Conclusions

Patients with NSCLC and physicians showed a strong preference for larger NGS panels such as CGP and WES/WGS, irrespective of high costs, when compared to other forms of genetic and genomic testing (small and medium panels or sequential tests). Patients with NSCLC and physicians showed strong preferences for genomic tests that provided actionable outcomes, were affordable, timely, and negated the need for additional biopsy. At a policy level, consideration should be given to broader reimbursed access to CGP in the Australian setting for both tissue and liquid biopsies, in order to ensure equity is achieved for all.

## Figures and Tables

**Figure 1 jpm-12-00879-f001:**
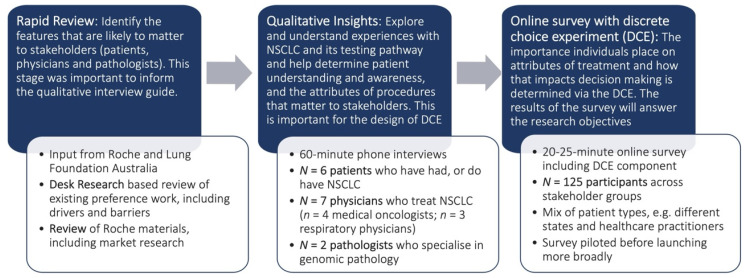
Overview of the methodological approach of this study. Non-small cell lung cancer (NSCLC); discrete choice experiment (DCE).

**Figure 2 jpm-12-00879-f002:**
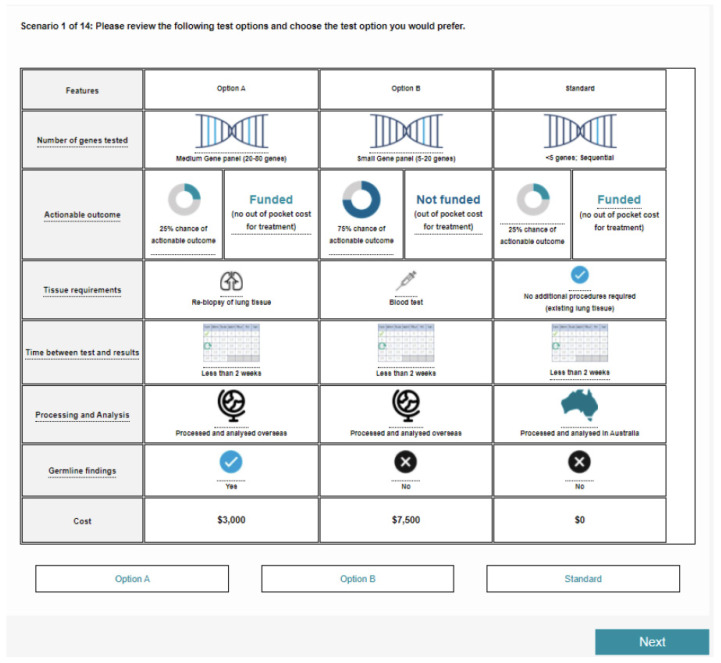
Example discrete choice experiment question.

**Figure 3 jpm-12-00879-f003:**
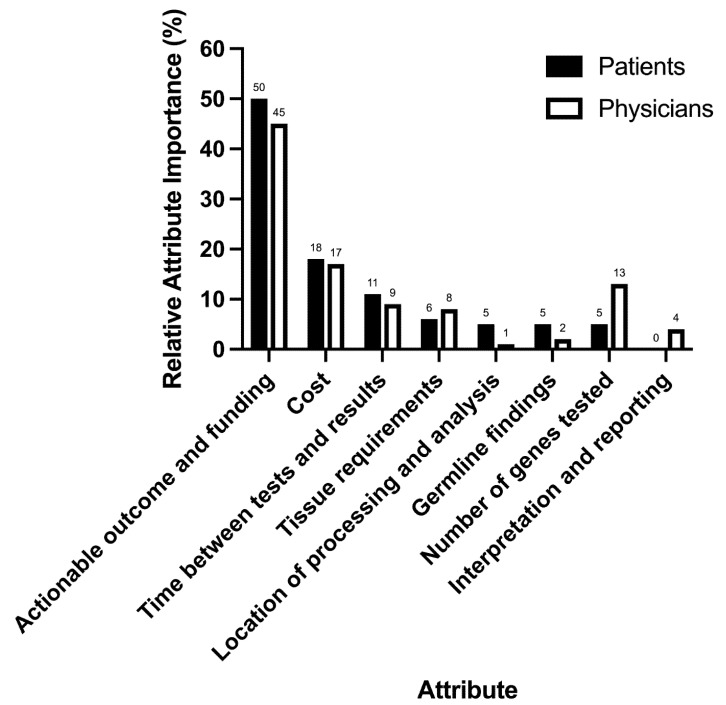
Relative attribute importance for patients and physicians.

**Table 1 jpm-12-00879-t001:** Discrete choice experiment attributes and levels.

Attribute Title Description in Patient Survey	Level Further Information (if Applicable)
Number of genes tested Number of genetic mutations tested for on the panel, some of which may have a potentially actionable outcome	Small gene panel (5–20 genes) 5–20 genetic mutations
	Medium gene panel (20–80 genes) 20–80 genetic mutations
	Comprehensive Gene panel (or CGP) (80–500 genes) 80–500 genetic mutations
	Whole Exome/Whole Genome sequencing (>500) More than 500 genetic mutations
Chance of actionable outcome Likelihood that the test will result in an actionable outcome for your treatment. e.g., targeted therapy; change/cessation of ineffective treatment; dosing requirements/adjustments	<1% chance <1% chance that the test may impact treatment decisions
	25% chance ** 25% chance that the test may impact treatment decisions
	50% chance 50% chance that the test may impact treatment decisions
	75% chance 75% chance that the test may impact treatment decisions
Funding for treatment Whether the actionable outcome for your treatment is funded (no out of pocket cost to you) or not funded (may be an out of pocket cost to you)	Funded ** No out of pocket costs for treatment. e.g., targeted therapy is covered by the PBS; clinical trial
	Not funded Potential out of pocket costs for treatment e.g., overseas clinical trials; medication not covered on PBS
Tissue requirements What kind of tissue is needed for the genetic mutation test, and whether other procedures are required	No additional procedures required ** No additional procedures required (existing lung tissue)
	Blood test
	Re-biopsy of lung tissue
Time between test and results Test turnaround: time to receive results that may impact clinical decisions	Less than 2 weeks **
	2–4 weeks
	4–6 weeks
	6–12 weeks
Location of processing and analysis Where the sample is processed and analysed. This includes tissue transport and storage, data analysis and interpretation of results for delivery	Australia **
	Overseas
Interpretation and reporting * Information included in the report that is returned, and what degree of interpretation it provides for assisting in clinical decisions. A full list of genetic mutations tested for, even when no alterations are found, will be delivered as an appendix in all tests	Reports on genetic mutation status only **
	Reports on genetic mutation status and eligibility for established targeted therapies
	Reports on genetic mutation status and eligibility for targeted therapies that are established or under investigation
	Reports on genetic mutation status and eligibility for targeted therapies that are established or under investigation. Access to local Decision Support Service ***
Germline findings Whether there is a 15% chance of detecting additional hereditary or inherited mutations. These findings do not guide treatment decisions, though may impact yours and your family decisions in other ways	Yes
	No **
Cost of test Out of pocket cost for the test	$0 **
	$1500
	$3000
	$4500
	$6000
	$7500

* Note: Shown to physicians only. Description is shown for physician survey. ** Attribute levels in the opt-out representing standard single-gene sequential testing. *** Decision Support Service assists with genomic interpretation, which may or may not incur additional costs, e.g., access to molecular tumour boards, expert case consultants and educational materials.

**Table 2 jpm-12-00879-t002:** Demographics of included participants in the discrete choice experiment.

Demographic	Patient Respondents (*n* = 45)	Clinician Respondents (*n* = 44)
	*n* (%)	*n* (%)
Gender identity		
Female	27 (60%)	9 (20%)
Male	18 (40%)	34 (77%)
Not answered		1 (2%)
Age		
31–40	10 (22%)	14 (32%)
41–50	12 (27%)	17 (39%)
51–60	7 (16%)	9 (20%)
61–70	12 (27%)	3 (7%)
71–80	4 (9%)	1 (2%)
Area of residence/practice		
Metropolitan/City	31 (69%)	
Regional	8 (18%)	
Rural	6 (13%)	
Stage at diagnosis		-
Early stage (I or II)	16 (36%)	
Advanced stage (III or IV)	26 (58%)	
Unknown/Not staged	3 (7%)	
Current stage		-
Early stage (I or II)	3 (7%)	
Advanced stage (III or IV)	36 (80%)	
Unknown/Not staged	6 (13%)	
Current line of therapy *		-
unknown	2 (4%)	
1	11 (24%)	
2	13 (29%)	
3	14 (31%)	
≥4	5 (11%)	
Length of time treating patients with lung cancer		
3–4 years		4 (9%)
5–6 years		5 (11%)
7–8 years		6 (14%)
9–10 years		3 (7%)
>10 years		26 (59%)
Specialty type		
Medical oncologist		23 (52%)
Respiratory physician		21 (48%)
Proportion of patients for whom genetic mutation testing for NSCLC has been ordered, mean (SD)		59% (36%)

Note: Percentages may not add up to 100% due to rounding error. * Line of therapy participant is currently on. NSCLC, non-small cell lung cancer; SD, standard deviation.

**Table 3 jpm-12-00879-t003:** Mixed multinomial logit model for patients.

Parameter	Symbol	Parameter		SE	T-Ratio
Random					
Number of genes tested					
Medium gene panel (20–80 genes)	𝛽_𝐺𝑀_	0.1311		0.183	0.71
Comprehensive gene panel (80–500 genes)	𝛽_𝐺𝐶_	0.2499		0.175	1.43
Whole exome/whole genome sequencing (More than 500 genes)	𝛽_𝐺𝑊_	0.2198		0.203	1.08
RC: Small Gene panel (5–20 genes)					
Tissue requirements					
Blood test required	𝛽_𝑇𝐵_	0.0089		0.128	0.07
Existing lung tissue (no procedures required)	𝛽_𝑇𝑁_	0.5523	***	0.18	3.08
RC: Re-biopsy of lung tissue					
Time between test and results					
Less than 2 weeks	𝛽_𝑇𝑀𝐸__2_	0.8012	***	0.225	3.57
2–4 weeks	𝛽_𝑇𝑀𝐸__4_	0.1570		0.185	0.85
4–6 weeks	𝛽_𝑇𝑀𝐸__6_	0.1693		0.176	0.96
RC: 6–12 weeks					
Cost of test					
Cost	𝛽_𝐶𝑂𝑆𝑇_	−0.0004	***	0	−5.66
Chance of actionable outcome and funding					
Actionable outcome: <1%, Funded	𝛽_𝐴𝐶__1__𝐹_	−2.3698	***	0.481	−4.93
Actionable outcome: 25%, Not funded	𝛽_𝐴𝐶__25__𝑁𝐹_	−1.1763	***	0.326	−3.61
Actionable outcome: 25%, Funded	𝛽_𝐴𝐶__25__𝐹_	−0.3719		0.263	−1.41
Actionable outcome: 50%, Not funded	𝛽_𝐴𝐶__50__𝑁𝐹_	1.6214	***	0.296	5.48
Actionable outcome: 50%, Funded	𝛽_𝐴𝐶__50__𝐹_	1.5471	***	0.299	5.17
Actionable outcome: 75%, Not funded	𝛽_𝐴𝐶__75__𝑁𝐹_	2.3385	***	0.424	5.51
Actionable outcome: 75%, Funded	𝛽_𝐴𝐶__75__𝐹_	3.7469	***	0.695	5.39
RC: Actionable outcome: <1%, Not funded					
Location of processing and analysis					
Australia	𝛽_𝐿𝑂𝐶_	0.4814	***	0.13	3.72
*RC: Overseas*					
Germline findings					
Yes	𝛽_𝐺𝐸𝑅𝑀_	0.4626	***	0.118	3.93
*RC: No*					
Non-random					
Constant					
Current testing scheme	CUR	−1.2161	***	0.448	−2.71

***-Significant at 1% level. Log-likelihood: −440.64056; Restricted log-likelihood: −593.25064; McFadden Pseudo R-squared: 0.2572438; Number of respondents: 45; Number of choice observations: 540.

**Table 4 jpm-12-00879-t004:** Mixed multinomial logit model for physicians.

Parameters	Symbol	Parameter		SE	T-ratio
Random					
Tissue requirements					
Blood test required	𝛽_𝑇𝐵_	0.1808		0.1382	1.31
Existing lung tissue (no procedures required)	𝛽_𝑇𝑁_	0.857	***	0.2353	3.64
RC: Re-biopsy of lung tissue					
Time between test and results					
Less than 2 weeks	𝛽_𝑇𝑀𝐸__2_	1.0958	***	0.309	3.55
2–4 weeks	𝛽_𝑇𝑀𝐸__4_	0.2182		0.1936	1.13
4–6 weeks	𝛽_𝑇𝑀𝐸__6_	−0.0401		0.2067	−0.19
RC: 6–12 weeks					
Cost of test					
Cost	𝛽_𝐶𝑂𝑆𝑇_	−0.0006	***	0.0001	−5.28
Chance of actionable outcome and funding					
Actionable outcome: <1%, Funded	𝛽_𝐴𝐶__1__𝐹_	−2.9017	***	0.6291	−4.61
Actionable outcome: 25%, Not funded	𝛽_𝐴𝐶__25__𝑁𝐹_	−1.777	***	0.4391	−4.05
Actionable outcome: 25%, Funded	𝛽_𝐴𝐶__25__𝐹_	−0.5473	*	0.3163	−1.73
Actionable outcome: 50%, Not funded	𝛽_𝐴𝐶__50__𝑁𝐹_	0.5834	**	0.2607	2.24
Actionable outcome: 50%, Funded	𝛽_𝐴𝐶__50__𝐹_	2.9653	***	0.476	6.23
Actionable outcome: 75%, Not funded	𝛽_𝐴𝐶__75__𝑁𝐹_	2.5349	***	0.4698	5.40
Actionable outcome: 75%, Funded	𝛽𝐴_𝐶__75__𝐹_	5.1731	***	0.8243	6.28
RC: Actionable outcome: <1%, Not funded					
*Non-random*					
Number of genes tested					
Medium gene panel (20–80 genes)	𝛽_𝐺𝑀_	0.7027	**	0.3316	2.12
Comprehensive gene panel (80–500 genes)	𝛽_𝐺𝐶_	0.6578	*	0.3837	1.71
Whole exome/whole genome sequencing (More than 500 genes)	𝛽_𝐺𝑊_	0.9801	**	0.4289	2.29
RC: Small Gene panel (5–20 genes)					
Location of processing and analysis					
Australia	𝛽_𝐿𝑂𝐶_	0.1712		0.1377	1.24
RC: Overseas					
Germline findings					
Yes	𝛽_𝐺𝐸𝑅𝑀_	0.2758	**	0.1333	2.07
RC: No					
Interpretation and reporting					
Reports on genetic mutation status and eligibility for established targeted therapies	𝛽_𝑅𝐸𝑃𝑇𝑇_	−0.2217		0.207	−1.07
Reports on genetic mutation status and eligibility for targeted therapies that are established or under investigation	𝛽_𝑅𝐸𝑃𝐼𝑁𝑉_	0.0613		0.213	0.29
Reports on genetic mutation status and eligibility for targeted therapies that are established or under investigation. Access to local Decision Support Service	𝛽_𝑅𝐸𝑃𝐷𝑆𝑆_	0.6043	**	0.281	2.15
*RC: Reports of genetic mutation status only*					
Constant					
Current testing scheme	CUR	−1.1812	**	0.5729	−2.06

RC: Reference category. * *p* < 0.05; ** *p* < 0.01; *** *p* < 0.0001.

**Table 5 jpm-12-00879-t005:** Total uptake of whole exome sequencing/whole genome sequencing or comprehensive genomic profiling by funding depending on tumour requirements.

	WES/WGS				CGP			
Funding of treatment	Funded		Not funded		Funded		Not funded	
	Physician	Patient	Physician	Patient	Physician	Patient	Physician	Patient
TOTAL UPTAKE If all re-tests are:								
Tumour re-biopsy	94.17%	87.17%	61.47%	64.10%	94.69%	87.11%	68.22%	87.90%
Blood test	96.98%	89.68%	70.26%	68.34%	97.26%	89.64%	77.04%	90.31%

Abbreviations: CGP = comprehensive genomic profiling; WES = whole exome sequencing; WGS = whole genome sequencing.

## Data Availability

The datasets generated and/or analysed during the current study are not publicly available due commercial sensitivities but may be available on reasonable requests; interested researchers should contact info@cappre.com.au.

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
