# Peer review of "Patient and Clinician Preferences for Genetic and Genomic Testing in Non-Small Cell Lung Cancer: A Discrete Choice Experiment"

_jpm, 2022, doi:10.3390/jpm12060879_

Round 1

Reviewer 1 Report

Fifer et al proposed a manuscript entitled “patient and clinician preferences for genetic and genomic testing in non-small cell lung cancer. A discrete choice experiment”

In general, the subject is timely, wondering what preoccupations both for patients and physicians in cases of genetics are testing in NSCLC context. As well introduced, international guidelines are published. However, clinical practice can be far from these recommendations as highlighted in Australia by authors. Methods and well described and data processing are adapted. Results are concordant with conclusions: patients and physicians   showed a strong preference for larger NGS panels with actionable drivers, irrespective of costs, mostly without additional biopsy and correct delay. A graphical abstract is really usefull for the comprehension of the study and improve the quality of the MS.

As minor limitations, I only suggest presenting:

  • Line 45 to 50 : current 1st line TKI for EGFR (Osimertinib), ALK (Alectinib, Brigatinib or Lorlatinib) and ROS1 (Crizotinib)
  • Introduction : Summary in a few words the differences between oncogenic driver targetable in first line (EGFR/ALK/ROS1) vs others in line 2+(KRAS G12C, MET, RET …)
  • Table 2 : Present in a continuous valor (median (range) or median (SD) for age at least and (if possible) length of time treating patients with lung cancer

After minor modifications, this manuscript can be considered for publication.

Thanks to the authors and editorial team for this review request.

Author Response

The authors thank the reviewer for their constructive feedback, which has improved the manuscript.

Reviewer 1:

In general, the subject is timely, wondering what preoccupations both for patients and physicians in cases of genetics are testing in NSCLC context. As well introduced, international guidelines are published. However, clinical practice can be far from these recommendations as highlighted in Australia by authors. Methods and well described and data processing are adapted. Results are concordant with conclusions: patients and physicians   showed a strong preference for larger NGS panels with actionable drivers, irrespective of costs, mostly without additional biopsy and correct delay. A graphical abstract is really usefull for the comprehension of the study and improve the quality of the MS.

As minor limitations, I only suggest presenting:

  • Line 45 to 50 : current 1stline TKI for EGFR (Osimertinib), ALK (Alectinib, Brigatinib or Lorlatinib) and ROS1 (Crizotinib)
  • Introduction : Summary in a few words the differences between oncogenic driver targetable in first line (EGFR/ALK/ROS1) vsothers in line 2+(KRAS G12C, MET, RET …)
  • Table 2 : Present in a continuous valor (median (range) or median (SD) for age at least and (if possible) length of time treating patients with lung cancer

After minor modifications, this manuscript can be considered for publication.

Thanks to the authors and editorial team for this review request.

Reviewer 1

Author Response

Line 45 to 50 : current 1st line TKI for EGFR (Osimertinib), ALK (Alectinib, Brigatinib or Lorlatinib) and ROS1 (Crizotinib)

Line 52- Amended to include 1st line therapies in Australia.

In Australia, the current 1st line TKI treatments for EGFR are erlotinib, gefitinib and afatinib (an osimertinib for EGFR T790M-positive); for ALK are alectinib ceritinib or crizotinib; and for ROS1 is crizotinib

Introduction : Summary in a few words the differences between oncogenic driver targetable in first line (EGFR/ALK/ROS1) vs others in line 2+(KRAS G12C, MET, RET …)

Amended line 57.

“Current standard of care diagnostic testing for NSCLC in Australia includes sequential single-gene testing for EGFR, ALK and ROS1, prioritised due to their population prevalence and subsidised associated treatments”

Table 2 : Present in a continuous valor (median (range) or median (SD) for age at least and (if possible) length of time treating patients with lung cancer

Only categorical data were collected and therefore this is unavailable.

Reviewer 2 Report

Thank you for the opportunity to review the manuscript.

This is the study to explore patient and clinician preference for genetic and genomic testing in non-small cell lung cancer using a discrete choice experiment in Australia. Authors developed the survey for DCE with stakeholders (patients, medical oncologists, respiratory physicians, and pathologists) and conducted DCE with patients and providers. The sample size is relatively small for DCE. The findings of this study showed the most important attribute for both patients and clinicians was the likelihood of an actionable test.

The research question is clinically relevant and important in health care policy. The findings of this study are important. However, there are major comments in the methods and discussion.

Major comments

  1. Despite the relatively small sample size and some challenges of DCE, what was the rationale to use DCE on this complex topic instead of other study methods, such as simple surveys and formative interviews?
  2. Lack of details in the recruitment methods for readers to evaluate for potential bias
  3. Although the authors acknowledged the potential selection bias in the discussion, discussion of the potential selection bias is limited. Patient participants of this study are much younger than the general lung cancer population. There is a limited report of patient socioeconomic characteristics and patient’s own experience of genetic and genomic testing and targeted therapy which could influence their decision and the study outcomes.
  4. The interview guide and details of qualitative analysis of the in-depth interview to identify attributes are missing although selecting attributes is one of the critical steps of DCE.
  5. The discussion currently has 11 paragraphs. The discussion section could be shortened and more focused. Discussion of the findings of this study compared with other studies in the field is limited. Some of the discussion points conflict with each other.

Minor comments

Abstract

  1. Line 20 “secondary findings”

What do the secondary findings mean?

Introduction

  1. Paragraph 1, lines 34 to 36

These lines are more about lung cancer in general, not specific for patients who could be a candidate for targeted therapy. It may be more important to describe the content more specifically for patients who could be a candidate for genetic and genomic testing and targeted treatment and the possible effect of targeted treatment in the introduction.

  1. Paragraph 2 describes the historical background of molecular testing and targeted therapy but does not describe the current practice around targeted therapy. I suggest updating this section to reflect the current practice.
  1. Paragraph 3, lines 51-52 “Current standard of care…ROS1”

Please insert the reference.

  1. Paragraph 3, lines 53-54 “high rate if repeat biopsy”

Do you have the actual number?

  1. Line 60 “GCP’

I think that this is the first time. Is it “GCP” or “CGP”? Please spell out.

  1. In DCE, the survey asks the number of genes tested “small,” “medium,” “comprehensive” gene panel, and ‘whole genomic sequencing” in table 1. Which option in DCE corresponds to “GCP”? Table 5 shows “WES/WCG” and “CGP.” In the discussion, only WES/WCG and GCP are mentioned.
  2. Is there any previous research on this topic? If so, please summarize the findings of the prior work in the introduction to emphasize the research question this study is addressing.

Methods

  1. The presentation of methods is a little difficult to follow. A Figure to describe the flow of the study may be helpful.
  2. 2.2 population

Please explain the reason for including only patients with stages IIIb and IV?  

Please explain the reason for including respiratory physicians.

  1. 2 “population”  

Suggest “interview participants” rather than “population.”

  1. Line 111 “ a panel company (EKas)” “Roche”

What do they mean? Please explain more so that readers can evaluate the possibility of selection bias and the potential for the conflict of interest.

What was the recruitment method, by e-mail invitation?

  1. Interview approach

“Interview questions are included in the Supplemental Text”

I do not think that the interview guide was included in the supplement. Please check.

  1. 4 “data processing.”

Suggest “data analysis.” Please provide more details of the qualitative analysis. How did authors develop a codebook? Was coding done by two coders?

  1. 6 Study sample

Suggest using “Survey participants.”

What was the recruitment method? What is a panel company?

  1. Table 1 “Decision Support Service …educational materials.”

Suggest moving this explanation of “decision support service” in the footprint of Table 1. Is DSS available to all in Australia? Or is this part of the service of the testing company? Is it covered by the insurance? Is the cost included in the cost of the test?

  1. Line 178 “If you consider the following two test options, which one would be more likely to provide you with actionable information?” This question was not clear.

Results

  1. 3.1.1 “respondents

Suggest using “participant characteristics.”

  1. 1.2 patients

Two quotes included are the quotes from the same patient participant. Do you have any other quotes from another patient participant to show the diverse input from patient participants?

  1. 1.3 “Doctor 2, Medical Oncologist, NSW.”

This was the only time the term “Doctor” was used instead of “physician” in this manuscript.

  1. 2 Results of the DCE, line 289

“44 were respiratory physicians,” I think this is 21, not 44. Please check.

  1. Table 2

Do you have the information on whether patient participants were tested for genetic and genomic testing?

What does “current line of therapy?” mean? This is not self-explanatory.

  1. Subtitle “3.2.1 patients” and “3.2.2 physicians”. Suggest changing the subtitles to reflect the content.
  1. 3.2.1 Line 298 “WES/WCG.”

This is the first time? Please spell out.

  1. Tables 3 and 4 “random” and “non-random”

Are they necessary information in these tables?

  1. Line 304 “whether the outcome would be funded”

Is it the outcome or the treatment?

  1. Table 4

Please explain the meanings of *, **, *** in the footprint of table 4.

  1. 3 Attribute importance

Suggest combining 3.31 and 3.3.2 in one section

Discussion

  1. The focus of the discussion is not clear. What is the most significant finding of this study compared with the findings from other studies? What is the gap this study is filling compared to other studies? Some of the contents in the discussion are the repetition of the content in the introduction. Some of the discussion is more general discussion rather than a discussion based on the findings of this study.
  2. Paragraph 2 "the most important attribute was the likelihood of an actional outcome"

The unactionable outcome could happen not only by insufficient tissue or by the process of sequential single-gene testing. There are many genetic and genomic testing results that do not lead to any clinical action simply because we do not have the treatments to target all the mutations. Too much testing could cause more unactionable outcomes and increase the cost. This is important to mention.

  1. Line 412 "by Bast et al. (2020)"

Please insert the reference.

I am not sure whether your study suggested: "patients would like to receive non-actionable results."

  1. Line 427 "however, it should be noted that many patients and some physicians do not understand… including participation a DCE survey"

Is this the finding from the interview or other studies? If so, please insert the reference.

  1. As authors described in 4 and the previous studies showed, this topic is complex, and the patient’s knowledge and understating could be limited. Three is potential effect of patient participant’s understanding and knowledge on the results of DCE. There was limited evaluation of patient participants’ understanding and knowledge in this study.
  1. Discussion of the selection bias is limited. The age of patient participants is much younger than the general lung cancer patients. Also, there was limited report of the patient's socioeconomic characteristics and experience of genetic and genomic testing and targeted therapy which could influence the result of DCE.

Supplement

Suggest adding some explanation to the tables in the supplement.

Author Response

The authors thank the reviewer for their constructive feedback, which has improved the manuscript.

Reviewer 2:

Thank you for the opportunity to review the manuscript.

This is the study to explore patient and clinician preference for genetic and genomic testing in non-small cell lung cancer using a discrete choice experiment in Australia. Authors developed the survey for DCE with stakeholders (patients, medical oncologists, respiratory physicians, and pathologists) and conducted DCE with patients and providers. The sample size is relatively small for DCE. The findings of this study showed the most important attribute for both patients and clinicians was the likelihood of an actionable test.

The research question is clinically relevant and important in health care policy. The findings of this study are important. However, there are major comments in the methods and discussion.

Major comments

  1. Despite the relatively small sample size and some challenges of DCE, what was the rationale to use DCE on this complex topic instead of other study methods, such as simple surveys and formative interviews?
  2. Lack of details in the recruitment methods for readers to evaluate for potential bias
  3. Although the authors acknowledged the potential selection bias in the discussion, discussion of the potential selection bias is limited. Patient participants of this study are much younger than the general lung cancer population. There is a limited report of patient socioeconomic characteristics and patient’s own experience of genetic and genomic testing and targeted therapy which could influence their decision and the study outcomes.
  4. The interview guide and details of qualitative analysis of the in-depth interview to identify attributes are missing although selecting attributes is one of the critical steps of DCE.
  5. The discussion currently has 11 paragraphs. The discussion section could be shortened and more focused. Discussion of the findings of this study compared with other studies in the field is limited. Some of the discussion points conflict with each other.

Minor comments

Abstract

  1. Line 20 “secondary findings”

What do the secondary findings mean? Has been corrected to “germline findings”

Introduction

  1. Paragraph 1, lines 34 to 36

These lines are more about lung cancer in general, not specific for patients who could be a candidate for targeted therapy. It may be more important to describe the content more specifically for patients who could be a candidate for genetic and genomic testing and targeted treatment and the possible effect of targeted treatment in the introduction. It has amended to include clinical practice guideline recommendations

  1. Paragraph 2 describes the historical background of molecular testing and targeted therapy but does not describe the current practice around targeted therapy. I suggest updating this section to reflect the current practice.

  1. Paragraph 3, lines 51-52 “Current standard of care…ROS1”

Please insert the reference. This has now been referenced.

  1. Paragraph 3, lines 53-54 “high rate if repeat biopsy”

Do you have the actual number? RO AC has included

  1. Line 60 “GCP’

I think that this is the first time. Is it “GCP” or “CGP”? Please spell out. Resolved

  1. In DCE, the survey asks the number of genes tested “small,” “medium,” “comprehensive” gene panel, and ‘whole genomic sequencing” in table 1. Which option in DCE corresponds to “GCP”? Table 5 shows “WES/WCG” and “CGP.” In the discussion, only WES/WCG and GCP are mentioned.

  1. Is there any previous research on this topic? If so, please summarize the findings of the prior work in the introduction to emphasize the research question this study is addressing.

Methods

  1. The presentation of methods is a little difficult to follow. A Figure to describe the flow of the study may be helpful. A new figure (Figure 1) has been added.

  1. 2.2 population

Please explain the reason for including only patients with stages IIIb and IV? Clarified

Please explain the reason for including respiratory physicians. Clarified

  1. 2 “population”  

Suggest “interview participants” rather than “population.” Amended

  1. Line 111 “ a panel company (EKas)” “Roche”

What do they mean? Please explain more so that readers can evaluate the possibility of selection bias and the potential for the conflict of interest.

  1. Interview approach

“Interview questions are included in the Supplemental Text. Included

  1. 4 “data processing.”

Suggest “data analysis.” Please provide more details of the qualitative analysis. How did authors develop a codebook? Was coding done by two coders? Included

  1. 6 Study sample

Suggest using “Survey participants.” Included

What was the recruitment method? What is a panel company?

  1. Table 1 “Decision Support Service …educational materials.”

Suggest moving this explanation of “decision support service” in the footprint of Table 1. Is DSS available to all in Australia? Or is this part of the service of the testing company? Is it covered by the insurance? Is the cost included in the cost of the test? Included

  1. Line 178 “If you consider the following two test options, which one would be more likely to provide you with actionable information?” This question was not clear. Explanation below

Results

  1. 3.1.1 “respondents

Suggest using “participant characteristics.” Amended

  1. 1.2 patients

Two quotes included are the quotes from the same patient participant.

  1. 1.3 “Doctor 2, Medical Oncologist, NSW.”

This was the only time the term “Doctor” was used instead of “physician” in this manuscript. Amended

  1. 2 Results of the DCE, line 289

“44 were respiratory physicians,” I think this is 21, not 44. Please check. Amended

  1. Table 2

Do you have the information on whether patient participants were tested for genetic and genomic testing? Please see below for explanation

What does “current line of therapy?” mean? This is not self-explanatory.

  1. Subtitle “3.2.1 patients” and “3.2.2 physicians”. Suggest changing the subtitles to reflect the content.

  1. 3.2.1 Line 298 “WES/WCG.”

This is the first time? Please spell out. Amended

  1. Tables 3 and 4 “random” and “non-random”

Are they necessary information in these tables? Please see below

  1. Line 304 “whether the outcome would be funded”

Is it the outcome or the treatment?

  1. Table 4

Please explain the meanings of *, **, *** in the footprint of table 4.

  1. 3 Attribute importance

Suggest combining 3.31 and 3.3.2 in one section Amended

Discussion

  1. The focus of the discussion is not clear. What is the most significant finding of this study compared with the findings from other studies? What is the gap this study is filling compared to other studies? Some of the contents in the discussion are the repetition of the content in the introduction. Some of the discussion is more general discussion rather than a discussion based on the findings of this study.
  2. Paragraph 2 "the most important attribute was the likelihood of an actional outcome"

The unactionable outcome could happen not only by insufficient tissue or by the process of sequential single-gene testing. There are many genetic and genomic testing results that do not lead to any clinical action simply because we do not have the treatments to target all the mutations. Too much testing could cause more unactionable outcomes and increase the cost. This is important to mention.

  1. Line 412 "by Bast et al. (2020)"

Please insert the reference.

I am not sure whether your study suggested: "patients would like to receive non-actionable results."

  1. Line 427 "however, it should be noted that many patients and some physicians do not understand… including participation a DCE survey"

Is this the finding from the interview or other studies? If so, please insert the reference.

  1. As authors described in 4 and the previous studies showed, this topic is complex, and the patient’s knowledge and understating could be limited. Three is potential effect of patient participant’s understanding and knowledge on the results of DCE. There was limited evaluation of patient participants’ understanding and knowledge in this study.
  1. Discussion of the selection bias is limited. The age of patient participants is much younger than the general lung cancer patients. Also, there was limited report of the patient's socioeconomic characteristics and experience of genetic and genomic testing and targeted therapy which could influence the result of DCE.

Supplement

Suggest adding some explanation to the tables in the supplement.

Reviewer 2

Author Response

Despite the relatively small sample size and some challenges of DCE, what was the rationale to use DCE on this complex topic instead of other study methods, such as simple surveys and formative interviews?

The DCE was used in conjunction with both stated survey questions and formative interviews.

DCEs are part of a quantitative technique for measuring elicited preferences, rather than stated. It poses significant benefit over stated preference surveys in many ways including:

-         Humans are often poor predictors of their own expected choice behaviour. Elicited preference using trade-off designs such as these are more reflective of real-world decision-making heuristics

-         All choices are hypothetical, and thus decisions can be elicited for scenarios that are not existent in the market yet

-         It allows a model of decision making systems to be built, whereby maximum utility for the participant cohort (in this case, patients and physicians) can be determined

-         Choices are not tied to one specific product, thereby remaining unbiased to brand positioning thus allowing for a controlled, randomised experimental design

-         As mentioned above, having an understanding of patient WTP for products (such as CGP) is a powerful tool for the patient voice, and communicates the value of the product to decision makers including payers, policymakers and healthcare professionals

Lack of details in the recruitment methods for readers to evaluate for potential bias

We have now  altered the statement by adding more information on recruitment method (line 181).

‘Convenience sampling was used to identify potential participants. Participants opted-in to the survey after hearing about it from the Lung Foundation Australia via email or social media, through third-party marketing research services companies (Pure Profile, Dynata, Stable research, Lucid or EKas), or by community recruitment through social media. Participants were compensated for their time: AUD$30 (patients) or AUD$140 (clinicians) for their participation.’

Although the authors acknowledged the potential selection bias in the discussion, discussion of the potential selection bias is limited. Patient participants of this study are much younger than the general lung cancer population. There is a limited report of patient socioeconomic characteristics and patient’s own experience of genetic and genomic testing and targeted therapy which could influence their decision and the study outcomes.

The interview guide and details of qualitative analysis of the in-depth interview to identify attributes are missing although selecting attributes is one of the critical steps of DCE.

Due to the sample size meaningful interpretations and conclusions on the socioeconomic and experience drivers on the findings could not be made.

We have now included the interview guide in the supplementary section.

The discussion currently has 11 paragraphs. The discussion section could be shortened and more focused. Discussion of the findings of this study compared with other studies in the field is limited. Some of the discussion points conflict with each other.

We understand that the discussion is long, but we believe that we would be doing a disservice to the complexities of this area by condensing the discussion. We are providing discussion on the attributes and downstream consequences of a test from results obtained from a choice experiment. Patient and physician perspectives are discussed, as well as the limitations and future directions.

Abstract: Line 20 “secondary findings”

What do the secondary findings mean?

Secondary findings defined as findings that are not actionable and outside of the variants being requested

ABSTRACT

Paragraph 1, lines 34 to 36

These lines are more about lung cancer in general, not specific for patients who could be a candidate for targeted therapy. It may be more important to describe the content more specifically for patients who could be a candidate for genetic and genomic testing and targeted treatment and the possible effect of targeted treatment in the introduction.

We have now clarified this throughout.

Paragraph 2 describes the historical background of molecular testing and targeted therapy but does not describe the current practice around targeted therapy. I suggest updating this section to reflect the current practice.

Current practice recommendations for targeted therapy have now been included as suggested by the reviewer.

Paragraph 3, lines 51-52 “Current standard of care…ROS1”

Please insert the reference.

References have been added

Paragraph 3, lines 53-54 “high rate if repeat biopsy”

Do you have the actual number?

30% per existing citation (now in line 61)

Line 60 “GCP’

I think that this is the first time. Is it “GCP” or “CGP”? Please spell out.

We have spelled this out now and checked the document to ensure the correct acronym (i.e. CGP) is used.

In DCE, the survey asks the number of genes tested “small,” “medium,” “comprehensive” gene panel, and ‘whole genomic sequencing” in table 1. Which option in DCE corresponds to “GCP”? Table 5 shows “WES/WCG” and “CGP.” In the discussion, only WES/WCG and GCP are mentioned.

The term comprehensive which was used in the DCE corresponds to CGP. This has been made clearer in the text of the manuscript. We have checked the document to ensure the correct acronym (i.e. CGP) is used.

Is there any previous research on this topic? If so, please summarize the findings of the prior work in the introduction to emphasize the research question this study is addressing.

Additional information has been added to the introduction that outlines previous patient preference research and which highlights that no patient preference research on diagnostics has been performed.

METHODS:

The presentation of methods is a little difficult to follow. A Figure to describe the flow of the study may be helpful.

We have now added a figure as suggested (Figure 1).

2.2 population

Please explain the reason for including only patients with stages IIIb and IV?  

Please explain the reason for including respiratory physicians.

Inclusion of patients with stages IIIb and IV in the study aligns to current clinical practice and clinical guidelines that recommend broad panel genomic testing for this patient population. 

In Australia, respiratory physicians are also able to request genomic testing and are critical decision-makers.

2 “population”  

Suggest “interview participants” rather than “population.”

Amended as per reviewer’s suggestion

Line 111 “ a panel company (EKas)” “Roche”

What do they mean? Please explain more so that readers can evaluate the possibility of selection bias and the potential for the conflict of interest.

What was the recruitment method, by e-mail invitation?

Amended to “third-party marketing research services company” and recruitment method (i.e. email and social media campaign) has now been stated.

Interview approach

“Interview questions are included in the Supplemental Text”

I do not think that the interview guide was included in the supplement. Please check.

All 3 interview guides have now been included as a supplementary document.

4 “data processing.”

Suggest “data analysis.” Please provide more details of the qualitative analysis. How did authors develop a codebook? Was coding done by two coders?

Amended as per reviewer’s suggestion (line 146).

“All interviews were audio recorded and transcribed. Interview data including transcriptions and analysis were managed in a specialised qualitative research software (NVivo). Once familiarised with the interview contents, broad themes and subsequent codes developed a preliminary code frame. Interview contents underwent thematic analysis using this code frame. Finalisation of the code frame and interview contents was reached upon discussion between the two coders working independently.”

6 Study sample

Suggest using “Survey participants.”

What was the recruitment method? What is a panel company?

Amended as per reviewer’s suggestion. Amended to “third-party marketing research services company” and recruitment method (i.e. email and social media campaign) has now been stated.

Table 1 “Decision Support Service …educational materials.”

Suggest moving this explanation of “decision support service” in the footprint of Table 1. Is DSS available to all in Australia? Or is this part of the service of the testing company? Is it covered by the insurance? Is the cost included in the cost of the test?

DSS are not necessarily available to all of Australia. Some testing companies do offer DSS for a fee as part of the services. This has been clarified further in the footnotes for the table.

Line 178 “If you consider the following two test options, which one would be more likely to provide you with actionable information?” This question was not clear.

Please note that this knowledge-check survey question was presented with context cues in the survey. In the survey, it immediately follows a background and explanation, and the question itself is shown with imagery consistent with the DCE. It is shown as an example only in the text of the document. We have now clarified this in text (line 216).

RESULTS:

3.1.1 “respondents

Suggest using “participant characteristics.”

Amended as per reviewer’s suggestion

1.2 patients

Two quotes included are the quotes from the same patient participant. Do you have any other quotes from another patient participant to show the diverse input from patient participants?

Please note that the two quotes included here are for two separate points. The first well summarised the issues with the current system, and the second spoke of the opportunities that CGP would provide. This patient participant was eloquently spoken, and so provided well-delivered insights in the patient voice, which others found challenging.

However, we have now added three additional quotes from 3 different patients to this section.

1.3 “Doctor 2, Medical Oncologist, NSW.”

This was the only time the term “Doctor” was used instead of “physician” in this manuscript.

Amended as per reviewer’s suggestion

2 Results of the DCE, line 289

“44 were respiratory physicians,” I think this is 21, not 44. Please check.

Corrected to 21 (line 268)

Table 2

Do you have the information on whether patient participants were tested for genetic and genomic testing?

What does “current line of therapy?” mean? This is not self-explanatory.

A footnote has been included to explain “Current line of therapy”

Subtitle “3.2.1 patients” and “3.2.2 physicians”. Suggest changing the subtitles to reflect the content.

Amended to “Mixed multinomial logit model results for….”

3.2.1 Line 298 “WES/WCG.”

This is the first time? Please spell out.

Amended as per reviewer’s suggestion

Tables 3 and 4 “random” and “non-random”

Are they necessary information in these tables?

These are an important part of the analysis in DCEs. It is information necessary for the coefficients in the model, and how they were modelled. Coefficients are treated as random in these models to allow for heterogeneity (variation) in preferences. If there is a lack of variation in preferences for a coefficient in the model, it is treated as non-random.

This is explained in section 2.8 Data analysis.

Line 304 “whether the outcome would be funded”

Is it the outcome or the treatment?

Outcome could include but is not limited to treatment. This has been made clearer in the manuscript.

  1. Table 4

Please explain the meanings of *, **, *** in the footprint of table 4.

These have now been explained in the footnotes

  1. 3 Attribute importance

Suggest combining 3.31 and 3.3.2 in one section

Amended as per reviewer’s suggestion

DISCUSSION:

The focus of the discussion is not clear. What is the most significant finding of this study compared with the findings from other studies? What is the gap this study is filling compared to other studies? Some of the contents in the discussion are the repetition of the content in the introduction. Some of the discussion is more general discussion rather than a discussion based on the findings of this study.

The discussion has been revised to provide a more in depth analysis and interpretation of the results and their significance relative to other studies and clinical practice.

Paragraph 2 "the most important attribute was the likelihood of an actional outcome"

The unactionable outcome could happen not only by insufficient tissue or by the process of sequential single-gene testing. There are many genetic and genomic testing results that do not lead to any clinical action simply because we do not have the treatments to target all the mutations. Too much testing could cause more unactionable outcomes and increase the cost. This is important to mention.

Further discussion on the complex interactions/relationship between actionability and cost has been included.

Line 412 "by Bast et al. (2020)"

Please insert the reference.

I am not sure whether your study suggested: "patients would like to receive non-actionable results."

Best S, Stark Z, Phillips P et al. Clinical genomic testing: what matters to key stakeholders?. European Journal of Human Genetics. 2020;28(7):866-873. doi:10.1038/s41431-020-0576-1

According to the model from the derived data, a high proportion of participants would have elected to receive genomic information, even when there is a less than 1% chance that it will lead to an actionable outcome.

We have now changed the wording in main text to make this clearer.

Line 427 "however, it should be noted that many patients and some physicians do not understand… including participation a DCE survey"

Is this the finding from the interview or other studies? If so, please insert the reference.

We have removed this sentence. This argument has been moved to the Limitations section:

“Further, participants were provided full background and explanation on all attributes and levels of the task, and thus education and understanding that underpin their choices may not be representative of the real-world setting. However, a shift towards patient empowerment over recent years includes the development of consumer education resources for products and technologies. In this way, the setting in which patients are briefed prior to making decisions may be more representative of the cohort of the near future.”

As authors described in 4 and the previous studies showed, this topic is complex, and the patient’s knowledge and understating could be limited. Three is potential effect of patient participant’s understanding and knowledge on the results of DCE. There was limited evaluation of patient participants’ understanding and knowledge in this study.

Patient understanding and knowledge was assessed and evaluated in the simple survey that accompanied the DCE task. However, the findings of this were out of scope for the current review.

Understanding and knowledge information from our cohort could be included, however without matched population data on this, it does not add much value.

Instead, we focussed on the importance of patient education in support of making informed decisions. When educated - as in the conditions of the task - patient participant choices are not always reflected by what physicians think they want.

The following has been included in section 2.7

“The survey instrument comprised of the DCE task, as well as non-experiment questionnaire sections assessing participant characteristics, test and treatment history and background knowledge and understanding of genomics in cancer care”

Discussion of the selection bias is limited. The age of patient participants is much younger than the general lung cancer patients. Also, there was limited report of the patient's socioeconomic characteristics and experience of genetic and genomic testing and targeted therapy which could influence the result of DCE.

This has now been expanded on in the limitations section.

SUPPLEMENT: Suggest adding some explanation to the tables in the supplement.

The explanation/notation to the tables in this section are provided in the main text.

Round 2

Reviewer 2 Report

Thank you for your response.  

Minor comments

  • Line 98 “Greature understanding” 

? Greater understanding?

  • Line 299 “Patient 2, Male, Vic” 

VIC

  • Section 3.2.2 – Inconsistent format

likelihood with out-of-pocket costs (5.1731). In the physician sample a test with no additional procedures was preferred to one that required a re-biopsy (0.857). A test turnaround time of less than 2 weeks was significantly preferred to 6-12 weeks (1.0958). Tests with germline findings were preferred to those without (0.2758) and a test sample processed and analysed in Australia was preferred to overseas (0.1712).”

(5.131) -(β=5.1731) ?

  • 3.3. Attribute importance

Line 374 and 382 “Figure 2” and actual “figure 2” – It is “figure 3” now.

Line 381 “number of genes tested (13%)” In Figure 2, it says 17%. Please double-check.

Author Response

Reviewer feedback and author responses:

Patient and clinician preferences for genetic and genomic testing in non-small cell lung cancer. A discrete choice experiment.

The authors thank the reviewer for their constructive feedback, which has improved the manuscript.

Reviewer:

Comments and Suggestions for Authors

Thank you for your response.  

Minor comments

Reviewer comment

Author response

Line 98 “Greature understanding” 

? Greater understanding?

Has been corrected to ‘greater’ (line 98).

Line 299 “Patient 2, Male, Vic” 

VIC

Has been corrected to ‘VIC’ (line 300).

Section 3.2.2 – Inconsistent format

likelihood with out-of-pocket costs (5.1731). In the physician sample a test with no additional procedures was preferred to one that required a re-biopsy (0.857). A test turnaround time of less than 2 weeks was significantly preferred to 6-12 weeks (1.0958). Tests with germline findings were preferred to those without (0.2758) and a test sample processed and analysed in Australia was preferred to overseas (0.1712).”

(5.131) -(β=5.1731) ?

Has been corrected in section 3.2.2.

3.3. Attribute importance

Line 374 and 382 “Figure 2” and actual “figure 2” – It is “figure 3” now.

Line 381 “number of genes tested (13%)” In Figure 2, it says 17%. Please double-check.

Figure 2 has now been changed to ‘figure 3’ in section 3.3, and the figure legend of figure 3.

 “number of genes tested was checked and we verify that the correct percentage is 13%. Hence, we have changed figure 3 to reflect this (from 17% to 13% for number of genes tested).

We have also completed a thorough check of the data to ensure that all reported data in the figures are correct and aligned with the manuscript.